# ESTIMATING LATENT REGULARIZATION PARAMETERS IN ILL-POSED PROBLEMS WITH SEMI-DEFINITE PROGRAMMING

## ABSTRACT

Tikhonov smoothing is often used in estimation problems in ill-posed settings. In a variety of applications ranging from human-computer interaction to model explainability, it is important to retroactively estimate smoothing parameters from an already trained model. We introduce an inverse regularization problem – one that infers latent smoothing hyperparameters obtained from a trained model and its dataset; and show a fast and effective solution using semi-definite programming. The algorithm directly exploits the stationarity conditions of Tikhonov models to jointly recover the model parameter's prior mean $\mu$ and Tikhonov precision matrix $T$ from observed optimum $\theta^*$. Our method formulates this as a multi-constraint least squares problem, providing a novel and interpretable approach. Empirically, our results show that our solution outperforms Bayesian approaches and solver-agnostic baselines on diverse benchmarks including diabetes, lung cancer, and California housing datasets.

## 1 INTRODUCTION

In machine learning, ill-posed problems arise when a model's solution is not unique, lacks stability, or cannot be reliably determined from the available data, often leading to poor generalization and overfitting, e.g., a linear regression with more parameters than examples. Regularization addresses ill-posed problems in machine learning by adding a constraint or penalty to the loss function, which discourages overly complex models. Tikhonov regularizers, also known as ridge regularizers in machine learning, are a form of regularization that stabilize ill-posed problems by adding a penalty term to the loss function based on the $\ell_2$ norm of the model parameters. In mathematical terms, consider the following linear model trained over a dataset $\{(x_i, y_i)\}_{i=1}^{N}$:

$$\theta^* = \arg\min_{\theta} \sum_{i=1}^{N} \phi(\theta^T x_i - y_i) + \lambda \|A(\theta - \mu)\|_2^2,$$

where $\phi$ is some convex and differentiable loss function and $T = A^T.A$ is the Tikhonov regularization matrix. The problem above is a $(T, \mu)$-regularized model, where these two hyperparameters aid in stabilizing the solution.

The Bayesian interpretation of this setup is the maximum a posteriori (MAP) estimate under a Gaussian prior on the model parameters, where the prior assumes $\theta^*$ is normally distributed around $\mu$ and with covariance $\Sigma = T^{-1} = (A^T A)^{-1}$ ($E[\theta^*] = \mu$ and $Cov[\theta^*] = \Sigma$). The regularization term corresponds to incorporating this prior belief into the training process, and thus, can be interpreted as an *inductive bias* or a prior belief in the probability of any certain final solution $\theta^*$.

Normally, $(T, \mu)$ are given and the goal is to solve for $\theta^*$. In a number of important settings, $(T, \mu)$ are *latent* – when some black-box modeling procedure produces a model with unknown or implicit regularization parameters. $(T, \mu)$ give us important insight into prior modeling beliefs. Therefore, retroactive inference of these parameters can be very valuable.

**Example 1** *Consider a panel of clinicians that are observing the results of a noisy clinical trial. They are given a scatter plot of dosage $x$ and a noisy self-reported indicator of patient well-being*

*y, and each is asked to draw a best-fit line relationship between the two based on their clinical experience. How do we characterize the latent prior belief in each clinician?*

**Example 2** *Consider an LLM performing in-context learning where it classifies new examples provided to it based on a few given examples. How do we characterize the inductive biases used to make these inferences from a small number of examples?*

**Example 3** *Consider a machine learning system where weights $\theta^*$ and a training dataset $X, Y$ are known, however the provenance of $\theta^*$ is lost. The machine learning engineer would like to determine what optimization hyperparameters were likely used to derive $\theta^*$ for future training.*

Accordingly, we present a new estimation problem called *Inverse Regularization*. Given a model $\theta^*$ solved for a known loss function $\phi$ and a dataset $X, Y$, we attempt to infer the latent $(T, \mu)$. For Example 1, this can be stated as "assuming the clinician are internally solving a least squares regression, what prior beliefs do they have about the relationship between $X$ and $Y$?". For Example 2, this might be stated as "assuming that the LLM roughly behaves like a logistic regression, what prior beliefs do they have about the parameters?". The inverse problem is a bit more complex than the forward problem that solves for the model parameters. We contribute a new algorithm that directly exploits the stationarity conditions to jointly recover the model parameter's prior mean $\mu$ and Tikhonov precision matrix $T$ from observed optimum $\theta^*$. Our method formulates this as a multi-constraint least squares problem, providing a novel and interpretable approach.

## 2 OUR CONTRIBUTION AND RELATED WORK

In many ML problems, the choice of the regularizer is decisive for both stability and interpretability of solutions. Classical Tikhonov methods assume a fixed structure for the precision matrix $T$ (e.g., identity, finite-difference Laplacian) and often set the prior mean $\mu = 0$. These assumptions are restrictive: they impose generic smoothness priors rather than data-adapted ones, and they fail to capture cases where meaningful prior information is embedded in a nonzero $\mu$. In applications such as imaging, biomedical signal analysis, and causal time-series modeling, this mismatch can lead to biased reconstructions or loss of critical structure. A principled way to *recover* both $T$ and $\mu$ directly from observations addresses this gap.

Prior work to learning regularization has adopted multiple approaches of inferring the Tikhonov matrix which typically differ from our setting. Alberti et al. (2021) learn an optimal Tikhonov operator through risk minimization. Noschese & Reichel (2012) poses inverse matrix problems whose solutions produce application-adapted Tikhonov matrices. Bilevel methods Kunisch & Pock (2013); De los Reyes et al. (2017); Ehrhardt et al. (2023); Gazzola & Gholami (2024) tune regularization parameters indirectly via an outer loss. Bayesian methods Knapik et al. (2011); Bouriga & Féron (2011); Cho et al. (2020) estimate the prior covariance through hierarchical modeling, but usually assume a fixed or zero mean. Graph-based approaches Dong et al. (2016); Pavez & Ortega (2016) impose structural constraints such as Laplacian precision, rather than solving directly for $T$.

Ng et al. (2000) introduce the ill-posed problem of Inverse Reinforcement Learning. They seek to infer the underlying reward function from the previously learned optimal policy. Their work shows how model behaviour can be explained through latent reward structures. Their resolution of non-uniqueness of solutions closely mimics the goal of inverse regularization.

In contrast, we formulate an *inverse regularization* problem that uses the stationarity condition

$$T(\theta^* - \mu) = b^*,$$

where $(\theta^*, b^*)$ are observed from the forward regularization problem. Our key contribution is to *jointly recover* the prior mean $\mu$ and the precision matrix $T$ by solving a multi-constraint least squares system, extended with machine learning models to handle diverse priors. This provides a novel, direct route to an interpretable estimation of both prior mean and precision from observed optima.

Solving for $T$ and $\mu$ is non-trivial. The stationarity condition provides only indirect information, and both $T$ and $\mu$ are high-dimensional unknowns. Naive estimation may result in non-positive-definite $T$, instability due to ill-conditioning, or overfitting when multiple solutions exist. Bayesian

approaches handle uncertainty but are computationally expensive and often assume restrictive priors. Bilevel and risk-minimization approaches require large amounts of training data and do not directly tie to the optimality equations. Our formulation addresses these challenges by casting the problem as a multi-constraint least squares system, leveraging observed optimum $(\theta^*, b^*)$ and machine learning models to stabilize recovery.

The ability to recover an interpretable prior mean and precision has broad implications. In imaging, this enables data-driven reconstruction methods that adapt to underlying structures while remaining explainable. In healthcare time-series, it provides a way to learn clinically meaningful prior baselines ($\mu$) and structured dependencies ($T$) from observed dynamics, improving both prediction and causal interpretability. In general scientific computing, recovering $T$ and $\mu$ offers a bridge between purely data-driven methods and mechanistic priors, allowing practitioners to integrate observed outcomes with interpretable regularizers. Thus, our framework not only advances the mathematical foundations of inverse regularization but also provides practical tools for high-impact domains.

## 3 PROBLEM STATEMENT

In this section, we describe the inverse regularization problem. We will focus on the least squares loss $\phi(y - \theta^T x) = \|y - \theta^T x\|_2^2$, but the techniques easily generalize to many other typically used loss functions including hinge loss, logistic loss, and $\ell_1$ losses as well.

The standard approach to Linear Regression is to use ordinary least squares (OLS). However, when the problem is ill-posed (no unique solution – none or many) we can employ regularization – an ML technique that allows for an optimal trade-off between model complexity (bias) and out-of-sample performance (variance). We focus on Ridge regularization and it's generalization – Tikhonov regularization.

Here, we solve a stabilized least-squares problem of the form:

$$\theta^* = \arg\min_\theta \sum_{i=1}^{N} \|\theta^T x_i - y_i\|_2^2 + \|\alpha I . \theta\|_2^2$$

This is done particularly when dealing with multi-collinearity across variables. This formulation can be generalized to:

$$\theta^* = \arg\min_\theta \sum_{i=1}^{N} \|\theta^T x_i - y_i\|_2^2 + \lambda \|A(\theta - \mu)\|_2^2,$$

The Tikhonov matrix $T = A^T . A$ is a precision matrix, which can be interpreted as defining a prior belief over model parameters $\theta$ under a Bayesian lens.

**Solving the Forward Problem**: Given data matrix $X \in \mathbb{R}^{n \times d}$, target feature $y \in \mathbb{R}^n$, the known regularization matrix $A \in \mathbb{R}^{p \times d}$ (and as a result, known Tikhonov matrix $T \in \mathbb{R}^{d \times d}$), and the prior model parameter mean $\mu \in \mathbb{R}^d$, we can obtain the optimal solution:

$$\theta^* = \arg\min_\theta \sum_{i=1}^{N} \|X\theta^T - y\|_2^2 + \lambda \|A(\theta - \mu)\|_2^2, \tag{1}$$

To solve for $\theta$, we take a partial derivative of equation 1 w.r.t. $\theta$. This gives us the first-order optimal linear system:

$$\nabla\theta = 2X^T(X\theta - y) + 2\lambda A^T A(\theta - \mu) = 0$$
$$\implies (X^T X + \lambda A^T A)\,\theta = X^T y + \lambda A^T A \mu \tag{2}$$

Upon defining $T = \lambda\, A^T A \in \mathbb{R}^{d \times d}$, we get:

$$(X^T X + T)\,\theta = X^T y + T\mu \tag{3}$$

The LHS $X^T X + T$ is invertible when $\lambda > 0$ and $A$ has full column rank Tibshirani. In this case, we get a unique solution:

$$\theta^* = (X^T X + T)^{-1}(X^T y + T\mu) \tag{4}$$

This could be achieved in two ways: via a closed-form solution approach or via an iterative algorithm like L-BFGS (quasi-Newton method) Berahas et al. (2016); Liu & Nocedal (1989) or SGD Amari (1993).

### 3.1 Inverse Regularization Problem

Now, we are set up to describe the inverse regularization problem. We are given a modeling problem defined by the following equation:

$$\theta^* = \arg\min_\theta \sum_{i=1}^N \|\theta^T x_i - y_i\|_2^2 + \lambda\|A(\theta - \mu)\|_2^2,$$

and we know $\theta^*, \{(x_i, y_i)\}_{i=1}^N$. The objective is to infer the most likely $T = A^T A$ and $\mu$ used to produce $\theta^*$ from the given data.

The interpretation of this problem is most clear in an agentic context. Consider a black-boxy agent that is only known to be applying a linear policy that is a function of state $x_i$. We know that the agent derived its policy from a dataset of observations $X, Y$, but do not know exactly how it arrived at the observed linear policy. Here, the inverse regularization problem infers the agent's latent optimization parameters $T, \mu$ that best explain the observed $\theta^*$ assuming optimal behavior.

**Q1. Does this apply to non-linear modeling?** If the agent's policy is known to be non-linear, we can apply the problem above over locally linear fragments of the state-space. In other words, we can resolve what the latent regularization parameters are in a local neighborhood.

**Q2. How does this relate to explainability?** In many explainable AI techniques, the goal is to present simple input-to-output relationships between examples and their predictions for human understanding. In contrast, inverse regularization links a model's behavior to its training data, namely, how strong the inductive bias is in the modeling procedure. A weaker inductive bias means that the model more directly optimizes a loss on the data, and a stronger bias means that the model is more closely centered around $\mu$ ignoring other evidence.

## 4 Solving Inverse Regularization

We now present our techniques for solving inverse regularization to estimate the model parameter mean and the Tikhonov precision matrix in the case of regression (which, in-turn, gives us the mean-centered covariance matrix). At a high-level, our approach inverts the regularization equations – given the observed optimal solution and gradients, we recover the latent hyperparameters that could have generated them. We present this as a constrained optimization problem where the precision matrix is positive semi-definite (corresponds to a valid covariance matrix), and we infer the mean in closed form. We propose two methods of solving inverse regularization:

1. **Biconvex SDP** – Trace-minimizing SDP with closed-form $\mu$ update. Here, we alternate between a T-step and a $\mu$-step for the solver until convergence.

2. **Diverse Priors** – Recovery using diverse priors on $\mu$. Here, we aim for recovery using d diverse priors $\mu_j$ to produce d constraints.

**Stationarity Conditions** For the regularized least-squares problem in equation 1, the first-order optimality (stationarity) condition is obtained in equation 2 by setting the gradient of the objective to zero. We rearrange equation 3 to obtain the following form:

$$T(\theta - \mu) = -X^T(X\theta - y)$$

For notational convenience, we define

$$b = T(\theta - \mu), \tag{5}$$

so that the stationarity condition in equation 3 is equivalently written as

$$b = -X^\top(X\theta - y). \tag{6}$$

Equation 6 highlights that the regularization term $T(\theta - \mu)$ can be interpreted as balancing the residual correlation $X^\top(X\theta - y)$, thereby enforcing a trade-off between data fit and prior structure. We elaborate our solution methods below.

## 4.1 METHOD 1: BICONVEX SDP

Our first proposed solution – **Biconvex SDP** is a bi-convex trace-minimizing semi-definite programming solver, that jointly solves for $T$ and $\mu$ in alternate steps. Our aim is to pick the lowest-trace PSD solution that fits the constraint, while allowing for slack (noise).

We set $\mu_{\text{est}}^0$ to a zero vector, providing a neutral prior from which the alternating updates refine $T$ and $\mu$. We alternate between the 2 steps below until we reach a point of convergence.

- T-step (convex): For fixed $\mu_{\text{est}}$ at the $k + 1^{\text{th}}$ step, solve:
$$T_{\text{est}}^{k+1} = \underset{T_{\text{est}}, \mu_{\text{est}}}{\arg\min} \, \text{trace}(T_{\text{est}}^k) + \rho ||T_{\text{est}}^k(\theta^* - \mu_{\text{est}}^k) - b^*||_2^2$$

- $\mu$-step (closed-form): For fixed $T_{\text{est}}$ at the $k + 1^{\text{th}}$ step, solve:
$$\mu_{\text{est}}^{k+1} = \underset{\mu_{\text{est}}}{\arg\min} \, ||T_{\text{est}}^{k+1}(\theta^* - \mu_{\text{est}}^k) - b^*||_2^2 + \gamma ||\mu_{\text{est}}^k||_2^2$$

While this method neatly solves a biconvex problem and respects stationarity constraints, it's solution tends to a low-rank, under-determined $T_{\text{est}}$ (often rank 1) that satisfies the constraint, that's not necessarily close to the ground-truth $T_{\text{true}}$. We thus, propose an alternate approach, which enhances the solution such that we're more correctly able to recover the original Tikhonov matrix and mean (model identifiability).

## 4.2 METHOD 2: DIVERSE PRIORS

The second approach - recovery using **Diverse Priors** on $\mu$ addresses the identifiability issue by introducing multiple, diverse prior means. With only one prior $\mu$, the stationarity condition does not uniquely determine $T$; many PSD solutions can satisfy a single constraint. To resolve this ambiguity and narrow down the scope, we choose a collection of priors $\{\mu_j\}_{j=1}^J$, each producing a different forward solution $\theta_j$ and corresponding residual vector $b_j$ $(J \geq d)$ from an independent stationarity equation. Stacking these relations yields a linear system of equations in $T$. When the shifts $\{\theta_j - \mu_j\}$ span $\mathbb{R}^d$, the precision matrix $T$ is uniquely identifiable in the noiseless case. This method formalizes the intuition that diverse priors provide multiple "views" of the same operator, enabling exact recovery where a single constraint is insufficient.

We start by choosing $J$ diverse priors $\{\mu_j\}_{j=1}^J$. For each $j$, we solve the forward problem to obtain $\theta_j$ and compute $b_j = -X^\top(X\theta_j - y)$. By stacking the constraints $T_{\text{obs}}(\theta_j - \mu_j) = b_j$, we define the block matrices:
$$V = \begin{bmatrix} \theta_1 - \mu_1 & \cdots & \theta_J - \mu_J \end{bmatrix} \in \mathbb{R}^{d \times J}, \qquad B = \begin{bmatrix} b_1 & \cdots & b_J \end{bmatrix} \in \mathbb{R}^{d \times J}.$$
Then the constraints can be expressed the matrix equation:
$$T_{\text{obs}} V = B. \tag{B0}$$
We use L-BFGS (Quasi-Newton Constrained Residual Minimization) and sum-of-squares loss so that the stationarity equation is exactly recovered. For exact recovery, if $\text{rank}(V) = d$ (thus $J \geq d$ and the columns of $V$ span $\mathbb{R}^d$), then
$$T_{\text{obs}} = B V^{-1} = B V^+,$$
where $V^{-1}$ applies when $V$ is invertible, and $V^+$ denotes the Moore–Penrose pseudo-inverse. In the noisy setting (model mismatch), we can estimate $T_{\text{obs}}$ via PSD-regularized least squares:
$$\min_{T_{\text{obs}} \succeq 0} ||T_{\text{obs}} V - B||_F^2 + \gamma ||T_{\text{obs}}||_F^2 + \alpha \, \text{tr}(T_{\text{obs}})$$
Dropping the PSD constraint yields the closed form
$$\tilde{T}_{\text{obs}} = B V^\top (V V^\top + \gamma I)^{-1}. \tag{B3}$$
Since numerical solvers and pseudo-inverses may introduce asymmetry or small negative eigenvalues, we enforce the structural constraint $T \succeq 0$ by projecting onto the PSD cone. This is achieved by eigen-decomposition and clipping:
$$T_{\text{obs}}^{\text{sym}} = \tfrac{1}{2}(\tilde{T}_{\text{obs}} + \tilde{T}_{\text{obs}}^\top), \qquad T_{\text{obs}}^{\text{hat}} = \Pi_{\succeq 0}(T_{\text{obs}}^{\text{sym}}),$$
where $\Pi_{\succeq 0}$ denotes projection to the PSD cone via eigenvalue thresholding. This guarantees that the recovered operator is a valid Tikhonov precision matrix.

# 5 EXPERIMENTS

In these experiments, we aim to verify the correctness of a reconstructed Tikhonov precision matrix and prior mean (via Inverse Regularization methods) by setting up a forward regularization process for regression a priori. We choose a fixed Tikhonov matrix and prior mean as the ground truth for the forward process, which we hope to reconstruct in the inverse process. We assume that the regularization matrix $A$ has full column rank, the Tikhonov precision matrix $T$ is a square matrix, and it isn't rank-deficient, as there are more observations than features in our datasets.

## 5.1 PIPELINE

**Phase 1: Ground Truth Generation** The ground truth Tikhonov precision $T_{\text{true}}$ and prior mean $\mu_{\text{true}}$ are sampled from a Normal-Wishart prior, the conjugate family for Gaussian mean and precision. Let $Q \in \mathbb{R}^{p \times d}$ be a reference operator and hyperparameters: ridge strength $\lambda > 0$, degrees of freedom $\nu_0 \geq d$, mean prior strength $\kappa_0 > 0$, and mean center $m_0 \in \mathbb{R}^d$. The Wishart scale is

$$\Sigma_0 = \tfrac{\lambda}{\nu_0} Q^\top Q,$$

so that $\mathbb{E}[T_{\text{true}}] = \lambda Q^\top Q$. We then draw

$$T_{\text{true}} \sim \mathcal{W}(\nu_0, \Sigma_0), \qquad \mu_{\text{true}}|T_{\text{true}} \sim \mathcal{N}\big(m_0, (\kappa_0 T_{\text{true}})^{-1}\big).$$

This ensures that $T_{\text{true}}$ is PSD and that $\mu_{\text{true}}$ has scale consistent with its precision, providing realistic ground truth for the setting.

**Phase 2: Forward Regularization** Here, the goal is to solve for optimum $\theta^*$. The Forward Regularization step can be achieved in two ways – via a closed-form solution approach (equation 1), or via training an ML model with custom penalty via OLS (ordinary least squares). We choose to iteratively train a Machine Learning regressor model to obtain the optimal solution (equation 3).

The regressor model $M$ takes as input a standardized dataset $\{X, Y\}$ which is split it into train and test sets: $\{X_{\text{train}}, y_{\text{train}}\}, \{X_{\text{test}}, y_{\text{test}}\}$, known Tikhonov matrix $T_{\text{true}} = \lambda A_{\text{true}}^T . A_{\text{true}} \in \mathbb{R}^d$, and apriori model parameter mean $\mu_{\text{true}}$. Post model learning from the training data, it outputs its optimal solution $\theta^*$.

**Phase 3: Inverse Regularization** After the forward regularization phase, we reconstruct an estimate of both the Tikhonov matrix $T_{\text{est}} = A_{\text{est}}^T . A_{\text{est}}$ and the prior mean $\mu_{\text{est}}$, given X, y and $\theta^*$.

The inverse regularizer solver method takes as input, the standardized train and test datasets $\{X_{\text{train}}, y_{\text{train}}\}, \{X_{\text{test}}, y_{\text{test}}\}$, the optimal solution $\theta^*$, and the stationarity constraints $b^*$. It's output is the estimated Tikhonov precision matrix $T_{\text{est}} = A_{\text{est}}^T . A_{\text{est}} \in \mathbb{R}^d$, and the prior mean $\mu_{\text{est}}$.

We employ the following set of 10 techniques to solve inverse regularization:

**(a) 5 baselines**: {Identity Ridge, Diagonal Ridge, PSD LS Ridge, Zero Mean Ridge, Random SPD}
**(b) 3 Bayesian benchmarks**: {MAP, MCMC, VI}
**(c) 2 proposed methods (ours)**: {Biconvex SDP, Diverse Priors}

## 5.2 METRICS OF EVALUATION

We present the following three metrics in our results: (i) the relative Frobenius error in the Tikhonov precision matrix – which measures how well the estimated precision matches the ground truth structure; (ii) the mean error – quantifying accuracy in recovering the prior mean; and (iii) the relative Frobenius error in the covariance matrix – which evaluates recovery of uncertainty structure. Together, these metrics highlight fidelity to the underlying operator, accuracy of the estimated mean, and uncertainty in calibration.

$$\text{Relative Frobenius error of } T: \quad \text{RelFrob}(T) = \frac{\|\hat{T} - T_{\text{true}}\|_F}{\|T_{\text{true}}\|_F}, \tag{7}$$

$$\text{RMSE of } \mu: \quad \text{RMSE}(\mu) = \sqrt{\tfrac{1}{d} \|\hat{\mu} - \mu_{\text{true}}\|_2^2}, \tag{8}$$

$$\text{Covariance error:} \quad \text{RelFrob}(\Sigma) = \frac{\|\hat{\Sigma} - \Sigma_{\text{true}}\|_F}{\|\Sigma_{\text{true}}\|_F}, \qquad \hat{\Sigma} = \hat{T}^{-1}. \tag{9}$$

## 5.3 BASELINES

We employ the following baselines to compare to our methods:

1. **Identity Ridge**: Assume $T = \alpha I_d$ and find $\alpha$ using grid search along with least squares in $\mu$. This captures the simplest isotropic Tikhonov prior.

2. **Diagonal Ridge**: Assume $T = diag(w_1, w_2, ..., w_d)$ and fit each $w_i$ independently from constraints. This allows for axis-aligned regularization, while still being simple and easy to interpret.

3. **PSD Least Squares Ridge**: Here, we solve directly for $T$ by least squares, and then the result is projected to the nearest PSD matrix.

4. **Zero Mean Ridge**: We fix the prior mean to 0 and only learn $T$ via ridge-regularized least squares.

5. **Random SPD**: We sample random SPD (symmetric positive-definite) matrices (Wishart draws) as priors. This serves as a sanity check against chance learning.

## 5.4 BENCHMARKS: CONSTRAINT-BASED BAYESIAN INVERSE REGULARIZATION

We create the following constraint-based Bayesian methods as benchmarks for solving Inverse Regularization:

1. **MAP (Deterministic)**: Maximum A Posteriori (MAP) estimation. We solve:
$$(\hat{\mu}, \hat{T}, \hat{\tau}) = \arg \min_{\mu, T \succeq 0, \tau > 0} -\log p(\mu, T, \tau \mid \theta^*, b).$$
This yields a single point estimate (instead of the full posterior) and corresponds to regularized least-squares with PSD projection – balancing likelihood with prior beliefs.

2. **MCMC (Stochastic)**: We draw samples from equation 22 using Hamiltonian Monte Carlo (NUTS). Posterior means $\mathbb{E}[\mu], \mathbb{E}[T]$ are used as estimates, and posterior credible intervals quantify uncertainty.

3. **VI (Variational):** Approximate the posterior by a tractable family $q(\mu, T, \tau)$ (here, mean-field Gaussian on unconstrained parameters via ADVI (Zhang & Chen (2022))). Estimates are posterior means under $q$, with uncertainty from the variational approximation.

The theoretical details behind these Bayesian methods can be found in the appendix.

## 5.5 DATASETS

We showcase our experiments on the following three datasets:

- Diabetes dataset: This dataset from the *sklearn* library consists of 500 individual data records and 10 fields. It is drawn from a 10-dimensional Gaussian and a linear Gaussian noise model. The goal is to predict the continuous diabetes progression indicator target feature. `https://scikit-learn.org/stable/modules/generated/sklearn.datasets.load_diabetes.html`

- Lung cancer dataset: This dataset is synthetically generated from the Asia lung cancer known causal graph. It consists of 1000 individual data records and 7 fields. The goal is to predict whether the patient has either of tuberculosis or lung cancer. `https://www.bnlearn.com/documentation/man/asia.html`

- California housing dataset: This dataset from the *sklearn* library consists of 21,000 individual data records and 8 fields. The goal is to predict the Median house value for California districts. `https://scikit-learn.org/stable/modules/generated/sklearn.datasets.fetch_california_housing.html`

The above collection of datasets allows us to evaluate the performance of our methods in diverse settings. This allows for our work to be easily generalizable to other datasets (different distributions, causal priors), and other ML prediction tasks like classification.

We defer further implementation details and theory to the appendix.

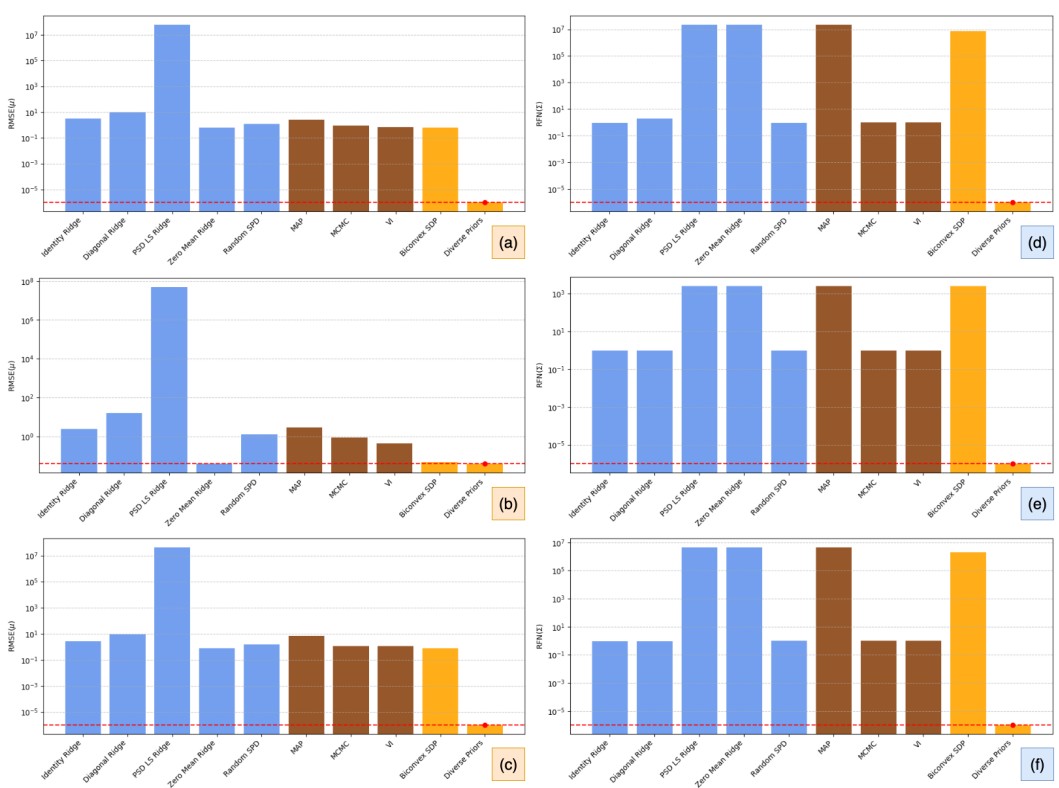

Figure 1: Left Column: Comparison of $RMSE(\mu)$ values and Right Column: Comparison of $RFN(\Sigma)$ across all techniques for the three datasets: (a),(d): Diabetes, (b),(e): Lung Cancer, and (c),(f): California Housing. The blue bars highlight the baselines, the brown ones - benchmarks, and the orange bars correspond to our methods.

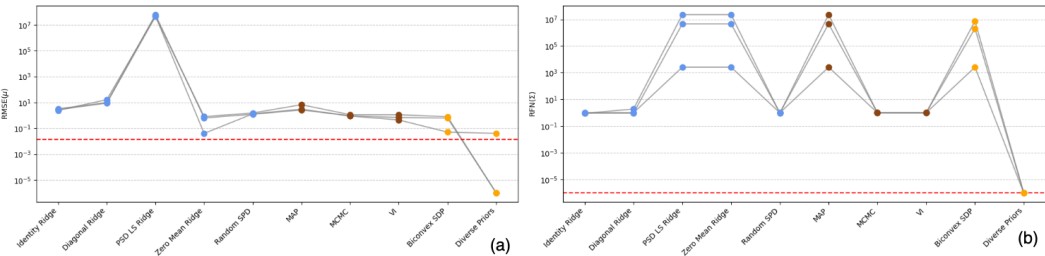

Figure 2: (a): Comparison of avg. $RMSE(\mu)$ values and (b): Comparison of avg. $RFN(\Sigma)$ values across all techniques averaged over all datasets.

## 5.6 RESULTS & ANALYSIS

We evaluate both the proposed solutions against baselines and benchmarks for key recovery metrics: RMSE of the recovered prior mean $\mu$ and Relative Frobenius Norm (RFN) error of the covariance matrix (inverse of the Tikhonov precision matrix). $\Sigma$. Figures 1 and 2 summarize the empirical findings. We show additional results on the RFN error of the recovered Tikhonov precision $T$ in the appendix. Figure 1 offers a comparison of the RMSE error of predicted means by the different methods for the three datasets in bar plots (a), (b), and (c) respectively (left column). It also shows a comparison of RFN error in the predicted covariance matrices in bar plots (d), (e), and (f) respectively (right column). Figure 2 offers a similar comparison – RMSE($\mu$) and RFN($\Sigma$), averaged across the three datasets - to account for generalization.

From these plots we can see that baselines based on classical ridge regularizers (Identity, Diagonal, PSD LS, and Zero Mean Ridge) and random SPD initializations perform poorly, often yielding errors several orders of magnitude larger than the ground truth. In particular, PSD LS ridge exhibits extreme instability, potentially because inversion of the ill-conditioned constraint matrix amplifies noise, resulting in large errors in both mean and covariance recovery. Zero-mean ridge, on the other hand, by fixing $\mu = 0$, forces the precision to absorb model mis-specification, making covariance estimates particularly unstable. MAP, MCMC, and variational inference (VI) achieve moderate performance, consistently outperforming naive baseline methods, but still incurring residual bias and variance. MAP underperforms MCMC and VI because it returns only a single mode of the posterior, ignoring uncertainty and multimodality. In contrast, MCMC and VI integrate over the posterior, yielding relatively more robust estimates of both the mean and covariance.

Now we discuss the results of our approaches. As expected, the **Biconvex SDP** method doesn't perform as well in recovering the covariance matrix because it prefers rank-deficient solutions. In contrast, the **Diverse Priors** method improves upon Bayesian benchmarks across all datasets by explicitly enforcing PSD structure and leveraging multiple independent constraints. It achieves near-exact recovery, with RMSE($\mu$) and RFN($\Sigma$) approaching numerical precision (dashed red line). This yields full-rank, well-conditioned estimates of both the mean and covariance, achieving the most accurate recovery overall. This validates the identifiability argument: multiple independent prior means provide sufficient constraints to uniquely recover the Tikhonov operator. Additionally, we observe the computational time for all methods. While all the baselines run in the order of a few milliseconds and our proposed methods – Biconvex SDP and Diverse priors run under 1 second, and the Bayesian approaches approximately require over a little over 1 hour to find a solution.

Overall, the results confirm that while traditional ridge and Bayesian approaches offer partial recovery, they cannot guarantee identifiability. Our proposed method based on Diverse Priors consistently outperforms alternatives, achieving exact recovery in the noiseless case and robust performance under perturbations. Our approach brings a nuance to testing the strength of priors that can benefit our understanding of ML models.

In the future, we plan on extending this work to classification models, and more smoothing priors – such as the Graph Laplacian for causal datasets, and other application-specific kernel based priors. We are also working on applying our work to enhance model explainability for tools like LIME Ribeiro et al. (2016).

## 6 CONCLUSION

We present a new estimation problem called Inverse Regularization that infers latent smoothing hyperparameters obtained from a trained prediction model and its dataset. We propose the novel "Diverse Priors" method for jointly inferring the model parameters' prior mean and Tikhonov precision matrix. This solution makes use of semi-definite programming and the Tikhonov model's stationarity conditions. We formulate it as a unique and interpretable multi-constraint least squares problem. Our proposed solution is fast and effective, and outperforms all baselines and Bayesian benchmarks presented on diverse datasets, delivering state-of-the-art recovery of the model parameters.

## 7 ETHICS STATEMENT

We confirm that our work adheres to ICLR's code of ethics. Our work does not involve human subjects, sensitive data, or applications that can cause societal harm. The datasets used (Diabetes, Asia lung cancer, and California housing) are publicly available and widely adopted benchmarks. Our methods are intended for improving understanding of regularization and inverse problems in machine learning optimization, and do not raise any ethical concerns.

We have made use of ChatGPT 5 specifically to aid with polishing writing at a grammatical level – sanity checks in sentence construction, and finding the right synonyms for words.

## 8 REPRODUCIBILITY STATEMENT

We ensure reproducibility of our results. All datasets are standard and publicly available. Complete algorithmic details, proofs, and hyperparameter settings are provided in the main text and appendix. In addition, all code used to run the experiments is included in the supplementary materials.

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

# A APPENDIX

We share some additional details, plots, results, and theory in the appendix.

## A.1 ASIA LUNG CANCER RESULT

Table 1: Error Metrics for Asia Lung Cancer Dataset

| Method | $\mathbf{RFN}(T)$ | $\mathbf{RMSE}(\mu)$ | $\mathbf{RFN}(\Sigma)$ |
|---|---|---|---|
| Identity Ridge | 0.92 | 2.40 | 1.00 |
| Diagonal Ridge | 0.99 | 16.38 | 1.00 |
| PSD LS Ridge | 0.94 | $4.91 \times 10^7$ | $2.61 \times 10^3$ |
| Zero Mean Ridge | 0.94 | 0.04 | $2.609 \times 10^3$ |
| Random SPD | 1.86 | 1.30 | 1.00 |
| MAP | 0.99 | 3.00 | $2.603 \times 10^3$ |
| MCMC | 3.08 | 0.88 | 1.00 |
| VI | 0.88 | 0.44 | 1.00 |
| Biconvex SDP | 0.86 | 0.05 | $2.608 \times 10^3$ |
| **Diverse Priors** | $1.0 \times 10^{-3}$ | 0.04 | $1.0 \times 10^{-3}$ |

## A.2 PIPELINE FLOWCHART

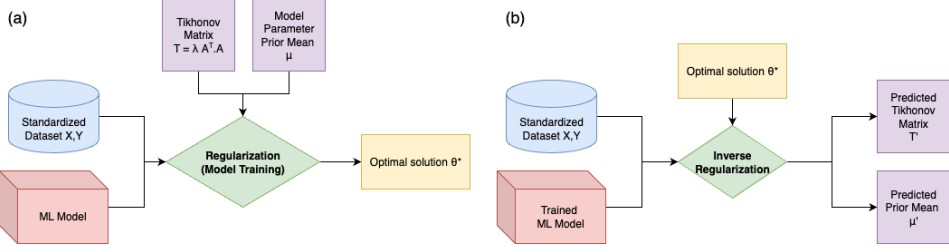

Figure 3: Pipeline of (a) forward regularization and (b) inverse regularization.

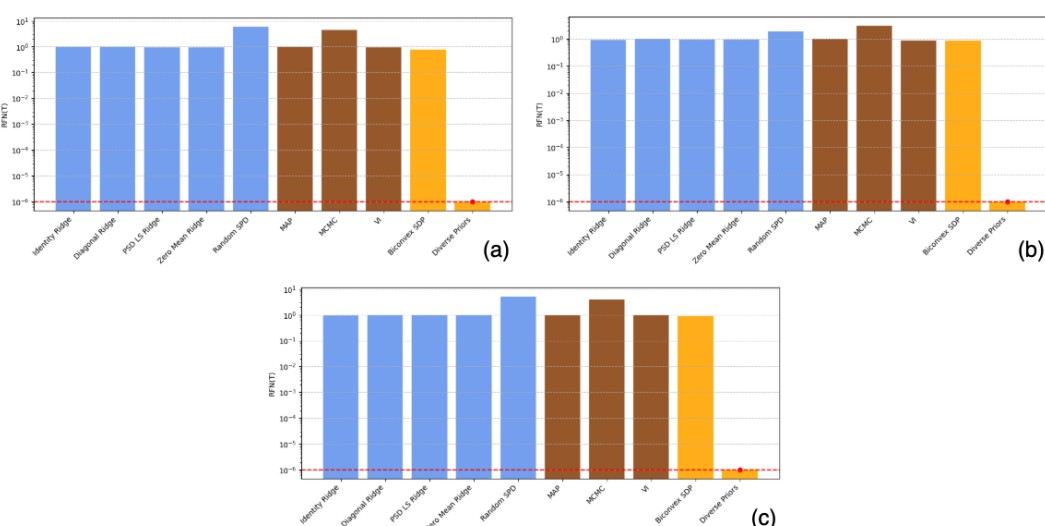

Figure 4: Comparison of $RFN(T)$ values across all techniques for the three datasets: (a) Diabetes, (b) Lung Cancer, and (c) California Housing.

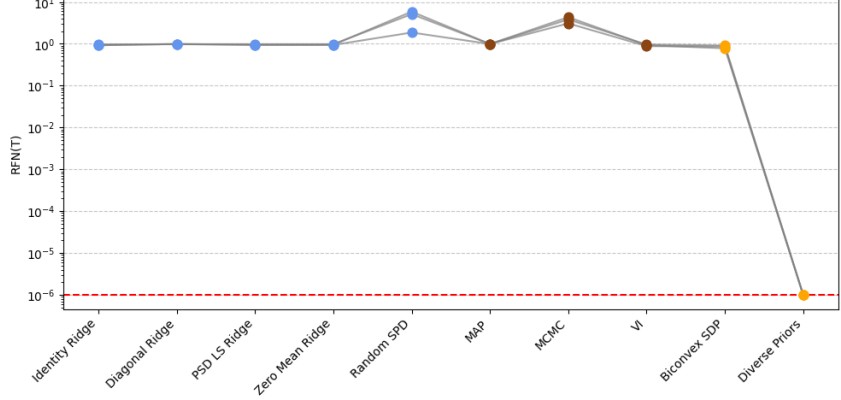

Figure 5: Comparison of $RFN(T)$ values across all techniques averaged over the three datasets.

### A.3 TIKHONOV PRECISION PLOTS

### A.4 PHASE 1: GROUND TRUTH GENERATION DETAILS

Fix $d$ and sample

$$\text{(i)} \quad \Sigma_{\text{true}} = AA^\top + \delta I_d, \ \delta > 0, \ A \in \mathbb{R}^{d \times d} \text{ with i.i.d. entries,} \tag{10}$$

$$\text{(ii)} \quad T_{\text{true}} = \Sigma_{\text{true}}^{-1}, \qquad \mu_{\text{true}} \sim \mathcal{N}(0, \sigma_\mu^2 I_d), \tag{11}$$

$$\text{(iii)} \quad \theta^* \sim \mathcal{N}(\mu_{\text{true}}, \Sigma_{\text{true}}), \tag{12}$$

$$\text{(iv)} \quad b = T_{\text{true}}(\theta^* - \mu_{\text{true}}) + \varepsilon, \quad \varepsilon \sim \mathcal{N}(0, \tau_{\text{sim}}^2 I_d). \tag{13}$$

### A.5 BENCHMARKS: CONSTRAINT-BASED BAYESIAN INVERSE REGULARIZATION

One approach to solving Inverse Regularization is to turn to Generative models like Bayesian Inference that assume Gaussian priors. For given $\theta^* \in \mathbb{R}^d$ obtained from a Tikhonov-regularized objective, let $b \in \mathbb{R}^d$ denote the stationarity right-hand side derived from data such that $b = -X^\top(X\theta^* - y)$. We aim to infer the prior mean $\mu \in \mathbb{R}^d$ and the Tikhonov (precision) matrix $T$ such that the (noisy) KKT stationarity holds.

The residual model can be defined as:

$$r \ = \ T(\theta^* - \mu) \ - \ b \in \mathbb{R}^d. \tag{14}$$

We assume an isotropic Gaussian noise model (with $\tau$ noise precision)

$$r \ \sim \ \mathcal{N}\left(0, \ \tau^2 I_d\right), \qquad \tau > 0. \tag{15}$$

Equivalently, the likelihood reads

$$p(b \mid \theta^*, \mu, T, \tau) \ \propto \ \exp\left(-\frac{1}{2\tau^2} \left\|T(\theta^* - \mu) - b\right\|_2^2\right). \tag{16}$$

We place independent priors on $\mu$, $T$, and $\tau$:

$$\mu \sim \mathcal{N}(\mu_0, \ \Lambda_0^{-1}), \tag{17}$$

$$T \sim \Pi_T, \qquad T \in \mathbb{S}_{\succeq 0}^d, \tag{18}$$

$$\tau \sim \Pi_\tau, \quad \text{e.g. } \tau \sim \text{HalfNormal}(\sigma_\tau). \tag{19}$$

Here, $\mu$ is the prior mean vector with hyperparameters $\mu_0$ is the prior mean center and $\Lambda_0$ is the prior precision of $\mu$, $\Pi_T$ is the prior on $T$, supported on $\mathbb{S}_{\succeq 0}^d$, and $\tau$ is the observation noise precision with prior $\Pi_\tau$ (e.g., HalfNormal with scale $\sigma_\tau$).

A convenient parameterization is via the covariance $\Sigma = T^{-1}$: we draw $\Sigma$ with an LKJ-Cholesky prior and set $T = \Sigma^{-1}$:

$$\Sigma \sim \text{LKJ}(\eta, \mathbf{s}), \quad T = \Sigma^{-1}, \quad \eta > 0, \ \mathbf{s} = \text{diag}(\sigma_1, \dots, \sigma_d). \tag{20}$$

With a single constraint (equation 14), the likelihood informs $T$ primarily along $v = \theta^* - \mu$.

At the optimizer $\theta^*$ of the Tikhonov-regularized objective $\mathcal{L}(\theta) = \|y - X\theta\|^2 + (\theta - \mu)^\top T(\theta - \mu)$, the stationarity condition reads

$$T(\theta^* - \mu) \ = \ -X^\top(X\theta^* - y). \tag{21}$$

We denote the right-hand side of equation 21 by $b \in \mathbb{R}^d$.

Given $(\theta^*, b)$, we place priors on $(\mu, T, \tau)$ and write the posterior as

$$p(\mu, T, \tau \mid \theta^*, b) \ \propto \ \exp\left(-\frac{1}{2\tau^2} \left\|T(\theta^* - \mu) - b\right\|_2^2\right) p(\mu)\, p(T)\, p(\tau). \tag{22}$$

We use weakly informative priors:

- $\mu_0 = 0$, $\Lambda_0 = \kappa_0 I_d$ (small $\kappa_0$),
- LKJ $\eta \in [1, 4]$ with scale vector $\mathbf{s}$
- and $\tau \sim \text{HalfNormal}(\sigma_\tau)$

### A.6 SOLUTION STABILITY

For numerical stability, one may model $\Sigma = AA^\top + \epsilon I_d$ with $\epsilon > 0$. To ensure feasibility, we explicitly constrain all recovered precision matrices to be positive semidefinite (PSD), which guarantees valid covariance estimates. Stability is promoted by incorporating Frobenius and trace regularization terms that prevent ill-conditioning and suppress extreme eigenvalues. Finally, to handle numerical degeneracies during matrix inversion or eigen decomposition, we add a small jitter term $\epsilon I$ (with $\epsilon > 0$) to the diagonal, ensuring strict positive definiteness and stable computations throughout the optimization process.

The entire code was written in Python 3.8 and is available in the supplementary files.

