# OpenReview forum: "Estimating Latent Regularization Parameters in Ill-Posed Problems with Semi-Definite Programming"
_ICLR.cc/2026/Conference — ICLR 2026 Conference Withdrawn Submission_

### Official Review · Reviewer_9rNt · 2025-10-26

**Soundness:** 1
**Presentation:** 2
**Contribution:** 1
**Rating:** 2
**Confidence:** 3

**Summary:**

The paper seeks to infer an L2/Tikhonov prior $(\mu,T)$ from a trained parameter $\hat\theta$ via the KKT relation
$X^\top(y-X\hat\theta)=\lambda\,T(\hat\theta-\mu)$. Two procedures are proposed: an alternating SDP and a stacked-equation ``Diverse Priors'' variant using multiple fits. While the algebraic setup is neat, the single-fit problem is non-identifiable in common regimes (large $n/p$, small $\lambda$), and the theory/experiments do not address this central limitation.

**Strengths:**

1. Clear algebraic formulation via KKT; easy to implement.
2. Two pragmatic solution paths (alternating SDP; stacked equations).

**Weaknesses:**

$\textbf{Non-identifiability}$ In $n \gg p$ or small $\lambda$ (typical weight decay), $X^\top(y-X\hat\theta)\approx 0$ so a single fit pins $\mu\approx\hat\theta$ but leaves $T$ unconstrained. This is not analyzed or bounded.

$\textbf{Scope shift.}$ The strongest results rely on multiple retrained solutions (``Diverse Priors''), which is a different setting than the advertised single-fit inverse problem.

$\textbf{Theory light.}$ No population target, no single-fit identifiability conditions, and no asymptotical or finite-sample guarantees; weaker than existing work on learning optimal Tikhonov regularizers (e.g., Alberti, Giovanni S., et al. "Learning the optimal Tikhonov regularizer for inverse problems." Advances in Neural Information Processing Systems 34 (2021): 25205-25216.).

$\textbf{Simple evaluation.}$ Lacks stress tests for small $\lambda$, large $n/p$, ill-conditioning, or modern training (decoupled weight decay); no simple diagnostics (e.g., whether $\rho=\|X^\top(y-X\hat\theta)\|_2$ is small).

**Questions:**

1. Under what minimal structure on $T$ (e.g., $T=\alpha I$) is single-fit recovery identifiable?

2. How does recovery error scale with $\lambda$ (or $\kappa$) and with the residual in realistic small-$\lambda$ regimes?

3. If multiple fits are essential, can guarantees be provided for that setting, with the single-fit scope clearly delimited?

---

### Official Review · Reviewer_sWgW · 2025-10-28

**Soundness:** 1
**Presentation:** 1
**Contribution:** 1
**Rating:** 0
**Confidence:** 4

**Summary:**

This paper investigates the "inverse" estimation of regularization parameters for linear problems (least-square with Tinkhonov regularization). It describes two methods for finding the mean and the matrix for regularizing the $\ell_2^2$ norm: one method relies on the first order conditions of the least-square, and the other one on an inverse problem.

**Strengths:**

The idea of finding regularization parameters from the data is an important problem.

**Weaknesses:**

I think the article suffers from many problems that prevents me to recommend it for acceptance: overall, the paper is poorly written (the claims are overly general, vague, and the same ideas are repeated multiple times) and lacks clarity, with a weak scientific contribution.

First the paper misses the context: the introduction contains no references and addresses a problem that has already been extensively studied since at least 30 years (the inverse estimation of regularization parameters).
The paper overlooks the existing literature entirely, mentioning only a few examples without providing any meaningful comparison or discussion with the proposed approach. The authors make no effort to place their work in a broader context: it remains unclear how this paper differs from existing approaches or what its real contribution is beyond what has already been published.

Moreover most of what is presented as a contribution could be directly taken from a standard machine learning textbook.
From a methodological point of view, there are many problems with the methods proposed. To take (a few) concrete exemples:

Method 1 is poorly defined and lacks consistency. For example, it is unclear what $b^\star$ represents or whether it depends on $T^k$. According to the text, either $b^\star = T^\star(\theta^\star - \mu^\star)$, which is not known, or $b^\star = T^k(\theta^\star - \mu^k)$, in which case the right-hand term disappears.
The statement that the solution is of rank one (and that a new “method” must therefore be proposed) suggests that the authors themselves did not clearly understand what they were doing.
In reality, this corresponds to the well-known minimization of the trace or nuclear norm of a matrix, a standard technique in inverse problems, which has no connection with the stationarity condition mentioned earlier. (As a side note: the derivation of the optimality conditions for least-square is well known and might as well be taken from a standard reference instead of being derivated).

Method 2 is no more convincing. It is never explained how the means $\mu_j$ are chosen in practice, and several sentences are nonsensical, such as: “We use L-BFGS (Quasi-Newton Constrained Residual Minimization) and sum-of-squares loss so that the stationarity equation is exactly recovered.”

The experimental section is a list of bullet points without clear discussion or critical analysis. Competing methods are not described, their relevance to the considered problem is not explained, and no theoretical comparison with the proposed approach is provided.
The reported results are doubtful: the evaluation metrics depend on $T_{\text{true}}$ for real datasets, a quantity that is in fact inaccessible. The authors mention that $T_{\text{true}}$ is sampled from a Wishart distribution based on a “reference operator” $Q$, but this $Q$ is never concretely defined for any real dataset.
Overall, the paper lacks rigor, novelty, and clarity. The paper fails to present any clear or original contribution to the field.

**Questions:**

See above

---

### Official Review · Reviewer_7iFP · 2025-10-31

**Soundness:** 1
**Presentation:** 1
**Contribution:** 1
**Rating:** 0
**Confidence:** 4

**Summary:**

The paper proposes a new setup: given a regularized Ridge regression estimation, i.e., given the learned parameters $\theta$, how to estimate the value of the Ridge regularization parameters?
For this new problem, authors derive a way to estimate the regularization parameters

**Strengths:**

I see no strength

**Weaknesses:**

- Lack of motivation. The authors introduce a whole new setup that lack true experimental grounding: **the authors exclusively study (generalized) linear regression**, but constantly provide motivating examples of much more complex settings, using vague and not precise formulation:
    - "Consider an LLM performing in-context learning where it classifies new examples provided to it based on a few given examples. How do we characterize the inductive biases used to make these inferences from a small number of examples?"
    - "“assuming that the LLM roughly behaves like a logistic regression, what
prior beliefs do they have about the parameters?”. "
    - "The interpretation of this problem is most clear in an agentic context. Consider a black-boxy agent that is only known to be applying a linear policy that is a function of state xi ."

    It feels like authors came with an artificial problem, and thus had to resolve to examples that are way to far from the setting they study.
    Could authors provide clear and impactful motivating examples for hte setting they study?

- IMO, Equation 1-4 are the basics of machine learning, and should be removed
- Can authors exactly pinpoint which optimization problem/fixed-point equation is solved?
- Could authors precisely show the proposed method in an algorithm environment? I did not understand the exact procedure.
- Can authors confirm that all experiments are done on "semi synthtetic" data?

**Questions:**

See weaknesses

---

### Official Review · Reviewer_HjYq · 2025-10-31

**Soundness:** 3
**Presentation:** 4
**Contribution:** 2
**Rating:** 4
**Confidence:** 3

**Summary:**

One can think of a least-squares problem as solving for a MLE problem for theta.  A ridge-regularized least-squares problem corresponds to having a Gaussian prior on the coefficients $\theta$.  This paper asks whether one can go backwards: Given some training data, and the value for $\theta$ that the algorithm produces can one infer the mean and variance of the prior (or equivalently the regularization parameters)?

The paper formulates this as an optimization problem and presents two methods for solving it.  The first is their “Biconvex SDP” method.  They let $T$ be the covariance matrix corresponding to the regularization prior, and $\mu$ be the mean.  They then solve for $T$ and $\mu$ by alternating between optimizing over $T$ and $\mu$.  Their optimization problem includes the nature term of the form $\|T(\theta - \mu) - b\|,$ corresponding to the fact that $\theta$ should fit the regularized problem, along with penalty terms that encourage $T$ to have small trace, and $\mu$ to have small norm.  $T$ is constrained to be PSD, and so their optimization procedure is implemented with SDP.

The second approach they call the “Diverse priors” method.  The issue is that $T$ is typically underconstrained, and so there is not generally a unique solution. To get around this, they imagine that there are several priors $\mu_1, \ldots, \mu_k$ that are chosen ahead of time. Then observing the coefficients $\theta_i$ produced for each $\mu_i$ gives more information about $T$, allowing it to be recovered.

**Strengths:**

Recovering the regularization parameters is a nice and well-motivated problem.  The authors give some examples where this might be useful.  For instance a model like an LLM might be used for some classification task, and it’s interesting to try to infer information about the model’s priors, which may give insight into the model.   The optimization methods are fairly standard, but natural and well-motivated. The experiments are generally reasonable and show that these algorithms can be implemented in practice.

**Weaknesses:**

The optimization methods are pretty standard, so in my opinion the main interest in the paper is in the idea of trying to recover priors from observing how a model behaves. This seems interesting, and I wish that the paper leaned into this a bit more.  As it stands I’m not entirely sold on the methods and experiments.  For instance, the experiments show that the diverse priors method dramatically outperforms everything else.  But if I understand the method  correctly, this isn’t particularly surprising. The diverse priors method gets to choose various $\mu_i$ and so this give the diverse priors additional information that the other algorithms don’t have.   It’s not clear to me whether one has the flexibility to choose $\mu$ in situations where this method would be useful.  The Biconvex SDP method also doesn’t seem to be a major improvement on other methods, given the experiments, although it appears to be comparable to more standard approaches.  This makes sense – the experiments do not seem to be designed specifically for low-rank $T$, and so the nuclear norm penalty might not give a particular advantage.

**Questions:**

Can the authors give an example of an application where the diverse priors methods could naturally be applied in practice?

What was the motivation for using a trace penalty rather than say a Frobenius penalty in the biconvex SDP method?

— Minor Comments —-
Page 1 “T = A^T. A” – This dot notation is used throughout.  I’m not sure if it’s clearer than just A^T A
“Black-boxy” -> “black box”
“While all the baselines…” – remove “while”

---

### Official Review · Reviewer_Qw1Q · 2025-11-01

**Soundness:** 1
**Presentation:** 1
**Contribution:** 1
**Rating:** 0
**Confidence:** 4

**Summary:**

This paper proposes methods for solving an inverse estimation problem, where given an optimal linear predictor $\theta^{\*}$ w.r.t the mean square loss, the goal is to infer the Tikhonov regularization parameter used in the forward problem (i.e., that which produces $\theta^{\*}$). The methods rely on the stationarity condition of the forward problem to recover the regularization parameters as the solutions to a constrained least squares objective. Experiments are conducted to quantify the performance of the methods against a collection of baselines.

**Strengths:**

The topic is relevant to the research community and has applications in high-stakes domains (e.g., medicine). The experiments are carried out on both real-world and synthetically generated data.

**Weaknesses:**

Unfortunately, the paper is ridden with shortcomings at present. Broadly speaking, significant parts of related literature are missing, the presentation lacks conciseness, the technical approach to solving the stated problem formulation is not well motivated or sound, and the paper contains technical mistakes. In particular:



1. **Related literature**
	* The claim of "we present a new estimation problem called Inverse Regularization." is not upheld in the light of i) missing a significant chunk of literature on inverse optimization (detailed below) and ii) a differential analysis w.r.t. the cited literature (given that the modeling approach is claimed to be novel, a thorough comparison of modeling approaches is needed to uphold the claim. The comparison should cover similarities in mathematical formulation, and a differential analysis of approaches to solving them w.r.t. existing methods).
	* The studied problem is part of the large niche of inverse optimization which is not discussed. Regularized problems have a corresponding constrained version (see [11] page 243). Through this lens, all works targeted at recovering constraints from the optimal model apply here, yet are not included/discussed (since the literature is too vast to cover here, please see [12, 13, 14] and backtrack through their cited references).


2. **Citation practice**
	* Inappropriate citations: The appropriate citation for SGD in line 161 is "Herbert Robbins and Sutton Monro (1951). A Stochastic Approximation Method”. The appropriate citation for L-BFGS line 161 besides the included Liu & Nocedal (1989) is "Nocedal, Jorge. "Updating quasi-Newton matrices with limited storage." Mathematics of computation 35.151 (1980)". Citing the sklearn library should follow the instructions [here](https://scikit-learn.org/stable/about.html), and cite "Pedregosa, Fabian, et al. Scikit-learn: Machine learning in Python. JMLR (2011)".
	* Missing citations:
		- Examples 1, 2 and 3 should be accompanied by citations and not be merely didactic. The reason is multifold: to inform the respective authors that new research has been developed for their real-world use case; to allow the reader to delve deeper into the practical problem, should they choose to; and to ground the statement that this problem is relevant in practice.
		- Section 4.1: The biconvex relaxation for approximately solving SDPs via alternate minimization was proposed, based on a summary search, by Shah, Sohil, et al. in "Biconvex relaxation for semidefinite programming in computer vision. ECCV. 2016.", yet this work is not cited.
		- Section 5.4: citation missing for Hamiltonian Monte Carlo, see [9-10] below


3. **Technical approach for solving the problem**
	* The method in lines 224 - 229 does not ensure PSD-ness of $T$ with projections, factorizations, or the like, so PSD-ness of the approximate solution is not guaranteed
	* Section 4.1: No motivation is provided for choosing the bi-convex approach over classical IPMs [1-3] or other cheaper first-order methods [4-8]. Moreover, no commentary is provided on the convergence problems encountered by this type of alternating minimization method and what, if anything, was done to address them (e.g., special initialization schemes).
	* Section 4.2: Solving the forward problem requires knowing $A$ and therefore $T$. However, by the problem statement, one does not have access to them, since they're precisely the parameters one seeks to recover. Furthermore, knowledge of $\mu_j$ is assumed in the forward problem. To my understanding, these statements are in contradiction.
     * Equation B3: the PSD constraint is dropped along with the Trace regularization term, and the latter is not justified
     * Line 264: Projecting the closed-form solution onto the PSD cone is not identical to computing the constrained minimizer.
     * The choice of SDP reformulation as a biconvex optimization problem is not motivated by the authors


4. **Inconsistencies and oddities in the mathematical notation and terminology**
	* Strange and non-standard mathematical notation:  (unless it is a repeated typo) line 040, 135, 142, 301, 307, 310 use notation $A^T.A$ to denote matrix multiplication. Even if this was intentional notation, it is neither consistent, nor defined e.g., in a notation section.
	* The authors define the regularization parameter $T = A^T.A$ in line 142, but later switch to $T = \lambda A^T A$ in line 154
	* The regularization strength is first denoted as $\alpha$ in line 135 but further renamed (without reason) as $\lambda$ in line 140
	* Once the data is placed in a matrix, the sum in line 148 should no longer be there
	* Notation in line 152, $\nabla \theta$ is improper. The gradient $\nabla$ should be applied to the $\theta$-dependent function inside the argmin of line 148
	* The SDP alluded to in section 4.1 has to be clearly stated in standard form, rather than described in words. Additionally, it is unusual to refer to it as a "Biconvex SDP" since the "biconvex" qualifier pertains to the specific manner of reformulating SDPs, rather than the SDPs themselves.

5. **Presentation**
	* The presentation is verbose, with sections being roughly repeated throughout the paper without adding further information or nuance (e.g., lines 64-73 and lines 113 - 120; lines 029-039 and lines 127-140; the forward optimization problem is stated 3 times without any significant variation in lines 148, 139, 167;)
	* Verbal description is preferred over concise mathematical formulations (detailed above), making this paper difficult to follow and imprecise.


6. **Experiments and figures**
	* The significance of the red dashed line is not explained, though one can presume it indicates the performance level of the best-performing approach
	* Convergence plots for the methods used in Sections 4.1 and 4.2 are not provided (and they should, in the light of above comments)
	* Hyperparameter tuning approach and final choices thereof are missing


7. **Other (minor) notation issues and typos**
	* Using $T$ for both the matrix transpose (as a superscript) and for denoting the regularization parameter makes the writing less neat. A solution would be to use the symbol $^\top$ to denote the transpose.
	* Line 437 $\Sigma$, line 191 "in-turn", line 130 "it's", 172 "boxy"




[1] Nesterov, Yurii, and Arkadii Nemirovskii. Interior-point polynomial algorithms in convex programming. Society for industrial and applied mathematics, 1994.

[2] Alizadeh, Farid. "Interior point methods in semidefinite programming with applications to combinatorial optimization." SIAM journal on Optimization 5.1 (1995): 13-51.

[3] Vandenberghe, Lieven, and Stephen Boyd. "Semidefinite programming." SIAM review 38.1 (1996): 49-95.

[4] Jaggi, Martin. "Revisiting Frank-Wolfe: Projection-free sparse convex optimization." International conference on machine learning. PMLR, 2013.

[5] Burer, Samuel, and Renato DC Monteiro. "A nonlinear programming algorithm for solving semidefinite programs via low-rank factorization." Mathematical programming 95.2 (2003): 329-357.

[6] Wen, Zaiwen, Donald Goldfarb, and Wotao Yin. "Alternating direction augmented Lagrangian methods for semidefinite programming." Mathematical Programming Computation 2.3 (2010): 203-230.

[7] Yang, Liuqin, Defeng Sun, and Kim-Chuan Toh. "SDPNAL+: a majorized semismooth Newton-CG augmented Lagrangian method for semidefinite programming with nonnegative constraints." Mathematical Programming Computation 7.3 (2015): 331-366.

[8] Yurtsever, Alp, Suvrit Sra, and Volkan Cevher. "Conditional gradient methods via stochastic path-integrated differential estimator." International conference on machine learning. PMLR, 2019.

[9] Duane, Simon, et al. "Hybrid monte carlo." Physics letters B 195.2 (1987): 216-222.

[10] Neal, Radford M. "MCMC using Hamiltonian dynamics." Handbook of markov chain monte carlo 2.11 (2011): 2.

[11] James, Gareth, et al. An introduction to statistical learning: with applications in R. Vol. 103. New York: springer, 2013.

[12] Ren, Ke, Peyman Mohajerin Esfahani, and Angelos Georghiou. "Inverse Optimization via Learning Feasible Regions." arXiv preprint arXiv:2505.15025 (2025).

[13] Chan, Timothy CY, and Neal Kaw. "Inverse optimization for the recovery of constraint parameters." European Journal of Operational Research 282.2 (2020): 415-427.

[14] Chan, Timothy CY, Rafid Mahmood, and Ian Yihang Zhu. "Inverse optimization: Theory and applications." Operations Research 73.2 (2025): 1046-1074.

**Questions:**

* Is there any reason why, in "Phase 2: Forward Regularization", you choose to train a model rather than rely on the closed-form solution?
* What initialization scheme do you use in the method in lines 224 - 229?
* How do the authors know that the poor performance of the "Biconvex SDP" is due to "preference for rank-deficient solutions" rather than the alternating minimization getting stuck in local minima?
* Figure 2a) would have benefited from "average + error-bar" type plots, as they convey the information more clearly.
* Do the authors have an explanation for the close performance of Zero Mean Ridge, and their two proposed approaches in Fig. 1b?

---

### Note · Authors · 2025-11-21

**Comment:**

We thank all the reviewers for their detailed and valuable feedback. We choose to withdraw our paper and focus on improving its issues.

**Withdrawal Confirmation:**

I have read and agree with the venue's withdrawal policy on behalf of myself and my co-authors.